# Metagenomic Insights into the Structure of Microbial Communities Involved in Nitrogen Cycling in Two Integrated Multitrophic Aquaculture (IMTA) Ponds

Qian Liu , Junnan Li, Hongwei Shan * and Yicheng Xie

The Key Laboratory of Mariculture, Ocean University of China, Ministry of Education, Qingdao 266003, China; liuqian9428@stu.ouc.edu.cn (Q.L.); lijunnan@stu.ouc.edu.cn (J.L.); xieyicheng1996@163.com (Y.X.)
* Correspondence: shanhongwei@ouc.edu.cn; Tel.: +86-532-82031912

**Abstract:** The microbial structure and metabolic potential, particularly with regard to nitrogen (N) cycling, in integrated multitrophic aquaculture (IMTA) ponds with shrimp remain unclear. In this study, an analysis of microbial community taxonomic diversity and a metagenomic analysis of N-related genes were performed in a shrimp-crab pond (*Penaeus japonicus-Portunus trituberculatus*, SC) and a shrimp-crab-clam pond (*P. japonicus-P. trituberculatus-Sinonovacula constricta*, SCC) to evaluate microbial structure and N transformation capacities in these two shrimp IMTA ponds. The composition of the microbial communities was similar between SC and SCC, but the water and sediments shared few common members in either pond. The relative abundances of N cycling genes were significantly higher in sediment than in water in both SC and SCC, except for assimilatory nitrate reduction genes. The main drivers of the differences in the relative abundances of N cycling genes in SC and SCC were salinity and pH in water and the $NO_2^-$ and $NH_4^+$ contents of pore water in sediment. These results indicate that the coculture of *S. constricta* in a shrimp-crab pond may result in decreased N cycling in sediment. The reduced N flux in the shrimp IMTA ponds primarily originates within the sediment, except for assimilatory nitrate reduction.

**Keywords:** nitrogen cycling; IMTA; *Penaeus japonicus*; *Portunus trituberculatus*; metagenomics

## 1. Introduction

Nitrogen (N) availability, cultured species performance, and biodiversity maintenance in aquaculture ecosystems are strongly regulated by microbes [1,2]. Additionally, microbial activity, diversity, and community compositions are strongly influenced by the cultured species, environmental properties, etc. [3,4]. Shrimp farming has been the fastest-growing sector of seawater aquaculture and has contributed to the supply of sufficient high-quality protein and improving economic benefits [5]. Given the high abundance, ubiquity, and important functions of microbes in aquaculture ecosystems, it is important to determine the main drivers of microbial patterns in these ecosystems. Studies have shown that the distribution patterns of microbial structures and functions in shrimp ponds are governed by one or multiple factors [6,7]. Some studies have indicated that temperature, pH, and N, or their interactions, mainly affect the microbial structure and function in some shrimp farming models [8,9], which hint at the optimal conditions for microbial structure and function and the ecological lifestyles of specific taxonomic communities in shrimp ponds.

The shrimp industry has experienced improvements and innovations, leading to many new farming modes, among which integrated multitrophic aquaculture (IMTA) including shrimp is the mode that has developed most rapidly along the northern and central coasts of China [5,10]. Shrimp IMTA is based on the multilevel resource utilization principle, according to which species from different trophic levels are introduced into shrimp ponds to create man-made ecosystems [11]. However, the responses of the microbial composition and functional communities in shrimp IMTA ponds may be different from those in traditional

shrimp farming ponds [11,12]. Thus, investigating the microbial structural and functional potential of shrimp IMTA ponds may provide evidence of the changes in targeted nutrient transformations in different types of shrimp ponds.

N is a key nutrient that may biologically limit production in aquaculture ecosystems, and microbes play a pivotal role in N cycling [13]. Six key processes are important in microbial-mediated N cycling including N fixation, assimilatory nitrate reduction, nitrification, denitrification, anaerobic ammonium oxidation (ANAMMOX), and dissimilatory nitrate reduction to ammonium (DNRA) [14,15]. Notably, denitrification, ANAMMOX, and DNRA have been viewed as important mechanisms of dissimilatory nitrate reduction in environments, and the level of sulfide (S) greatly influences the predominance of these processes [16–18]. Denitrification and sufficient organic carbon (C) levels favor the occurrence of heterotrophic denitrification, and sulfur can be utilized as an electron donor for $NO_3^-$ reduction via S-dependent autotrophic denitrification [19,20]. ANAMMOX involves the anoxic oxidation of ammonium utilizing $NO_3^-$ or $NO_2^-$ as an electron acceptor, and S-dependent ANAMMOX involves the anoxic oxidation of ammonium utilizing sulfate as an electron acceptor [21,22]. One form of DNRA involves fermentation, in which nitrate is reduced to $NH_4^+$ by the electron flow from organic C, while the other form, linked to S oxidation, is referred to as respiratory DNRA [23,24].

Environmental factors such as ammonium, temperature, pH, and microbial utilization can effectively change the amount of N cycling in ecosystems [25–29]. The specific genes involved in N cycling and the microbes that contain these genes play a vital role in N transformation [30]. However, the responses of microbial structure and potential functions to different types of shrimp IMTA ponds remain unknown. Thus, it is fundamental to identify the microbial community structure and the genes involved in N cycling processes, along with the differences in their relative abundance among different types of shrimp IMTA ponds.

In the present study, typo microbiome diversity and metagenomic analysis approaches were used to compare pond microbial structure and genes involved in N cycling in two different types of shrimp IMTA ponds, and the corresponding responses to water and sediment properties were investigated. Two typical types of shrimp IMTA ponds including a shrimp-crab pond containing *P. japonicus-P. trituberculatus* (SC) and a shrimp-crab-clam pond containing *P. japonicus-P. trituberculatus-S. constricta* (SCC) were chosen for this work. The taxonomic composition and β-diversity were determined to reflect the microbial structure, and microbial N cycling genes were identified to reflect the microbial potential of pond N decomposition and biosynthesis. The aims were to identify (1) the microbial structure and functions, especially regarding N cycling genes as well as compare the differences between the two shrimp IMTA ponds and between water and sediment; and (2) the environmental properties that affect the differences in the relative abundance of N cycling genes.

## 2. Materials and Methods

### 2.1. Experimental Design and Sampling

The ponds were located in Changbai Bay, Zhoushan City, Zhejiang Province, China, and the field work was carried out in August 2020. The experimental ponds and sampling site details are presented in Figure S1. Two culture ponds were designed (1.33 ha/pond), one of which contained a bispecies polyculture ecosystem consisting of *P. japonicus* and *P. trituberculatus* (SC), while the other contained a trispecies polyculture ecosystem consisting of *P. japonicus*, *P. trituberculatus*, and *S. constricta* (SCC). The details of the cultured species in SC and SCC are presented in Table S1.

In each pond, three sampling sites were established for the collection of three replicates. At each sampling site, water and sediment (0–5 cm) were collected. The water samples were filtered with 0.22 μm mixing fibroid membranes for microbial sampling, after which the microbe samples were frozen in liquid N and stored at −80 °C for DNA extraction, and the filtered water was stored at −20 °C for physiochemical analysis. For the sediment

samples, 10 g was weighed, frozen in liquid N, and then stored at $-80\,°C$ for microbial analysis, and the remainder was frozen ($-20\,°C$) for physiochemical measurements.

## 2.2. Measurement of Water and Sediment Physiochemical Indexes

Water temperature, dissolved oxygen (DO) concentrations, and salinity were determined with a portable fluorescence DO meter (YSI ProODO), and pH was determined with a pH meter (828+) [31]. The ammonia nitrogen ($NH_4^+$) concentration, nitrate ($NO_3^-$) concentration, and active phosphate ($PO_4^{3-}$) concentration were determined with a nutrient analyzer (DeChem-Tech, Hamburg, Germany), the nitrite nitrogen ($NO_2^-$) concentration was determined via the diazo coupling method, and the total nitrogen (TN) concentration and total phosphorus (TP) concentration were measured via the phosphorus molybdenum blue method [32].

The temperature, pH, and oxidation–reduction potential (ORP) of the sediment were determined using a pH/mV meter (IQ150, Spectrum) [23]. The sediment TN concentration and total carbon (TC) concentration were measured with an elemental analyzer (Vario ELIII, Hamburg, Germany). The pore water of the sediments was centrifuged at $-4\,°C$ for 15 min at 5000 r/min, and the supernatant was filtered with a 0.45 μm filter membrane, after which the $NH_4^+$ concentration and $NO_2^-$ concentration (P_$NH_4^+$ and P_$NO_2^-$) were determined according to Tu et al. and Laskov et al. [32,33]. Sediment $NH_4^+$ and $NO_2^-$ were extracted with 2 mol/L KCl and quantified by using a colorimeter [24].

## 2.3. Microbial DNA Extraction

Water and sediment microbial DNA were extracted using the E FastDNA® Spin Kit (Omega Bio-tek, Norcross, GA, USA) according to the manufacturer's instructions. The concentration and purity of the extracted DNA were determined with TBS-380 and NanoDrop2000 instruments, respectively. DNA extract quality was checked on a 1% agarose gel.

## 2.4. Illumina 16S rRNA Gene Sequencing Analysis

The V3-V4 hypervariable region of the bacterial 16S rRNA gene was amplified with the primer pair 338F (5′-ACTCCTACGGGAGGCAGCAG-3′) and 806R (5′-GGACTACHVGGG TWTCTAAT-3′) using an ABI GeneAmp® 9700 PCR thermocycler (ABI, Waltham, CA, USA). The PCR amplification of the 16S rRNA gene was performed with the PCR mixtures shown in Tables S2 and S3. The PCR products were extracted from a 2% agarose gel for purification and quantification using the AxyPrep DNA Gel Extraction Kit (Axygen Biosciences, Union City, CA, USA) and a Quantus™ Fluorometer (Promega, WI, USA), respectively. Based on the standard protocols of Majorbio Bio-Pharm Technology Co. Ltd. (Shanghai, China) for sequencing, the purified amplicons were pooled in equimolar amounts and subjected to paired-end sequencing on an Illumina MiSeq PE300 platform/NovaSeq PE250 platform (Illumina, San Diego, CA, USA). All sequences were deposited in the NCBI Sequence Read Archive (SRA) database under accession numbers SRR14203786, SRR14203784, SRR14203782, SRR14203792, SRR14203789, and SRR14203787 for the water samples and SRR15517008-SRR15517013 for the sediment samples.

## 2.5. Metagenome Sequencing, Data Processing, and Assembly

The extracted DNA was fragmented to an average size of approximately 400 bp using Covaris M220 (Gene Company Limited, Hong Kong, China) for paired-end library construction. A paired-end library was constructed using NEXTFLEX® Rapid DNA-Seq (Bioo Scientific, Austin, TX, USA). Adapters containing the full complement of sequencing primer hybridization sites were ligated to the blunt ends of the fragments. Paired-end sequencing was performed on the Illumina NovaSeq platform (Illumina Inc., San Diego, CA, USA) at Majorbio Bio-Pharm Technology Co. Ltd. (Shanghai, China) using NovaSeq Reagent Kits according to the manufacturer's instructions (www.illumina.com, accessed on: 18 February 2021). All sequences were deposited in the NCBI Short Read Archive

database (accession numbers: SRR15167360-SRR15167363, SRR15167366, and SRR15167367 for the water samples; SRR15167356- SRR15167359, SRR15167364, and SRR15167365 for the sediment samples).

The data were analyzed on the Majorbio Cloud Platform (www.majorbio.com, accessed on: 20 February 2021). The paired-end Illumina reads were trimmed of adaptors, and low-quality reads (length <50 bp or with a quality value <20 or having N bases) were removed by fastp (version 0.20.0) [34]. Bruijn graphs were used to assemble metagenomics data (MEGAHIT, version 1.1.2) [35]. The final assembled contigs (300 bp or longer) were selected for further analysis.

MetaGene (http://metagene.cb.k.u-tokyo.ac.jp/, accessed on: 20 February 2021) was used to predict open reading frames (ORFs) from each assembled contig [36]. The NCBI translation table was used to retrieve and translate the predicted ORFs (length being or over 100 bp) into amino acid sequences. CD-HIT (http://www.bioinformatics.org/cd-hit/, accessed on: 25 February 2021, version 4.6.1) [37] was used to construct a nonredundant gene catalog with 90% sequence identity and 90% coverage. After quality control, the reads were mapped to the nonredundant gene catalog (95% identity) by using SOAPaligner (http://soap.genomics.org.cn/, accessed on: 29 February 2021, version 2.21) [38], and gene abundance in each sample was evaluated.

Diamond (http://www.diamondsearch.org/index.php, accessed on: 15 March 2021, version 0.8.35) was used to align the representative sequences of the nonredundant gene catalog to the NCBI NR database (e-value cutoff of $1 \times 10^{-5}$ using) for taxonomic annotations [39]. Clusters of Orthologous Groups of Proteins (COG) annotation of representative sequences and Kyoto Encyclopedia of Genes and Genomes (KEGG) annotation (an e-value cutoff of $1 \times 10^{-5}$ were performed using Diamond (http://www.diamondsearch.org/index.php, accessed on: 18 April 2021, version 0.8.35) [39] against the eggNOG database and the KEGG database (http://www.genome.jp/keeg/, accessed on: 19 April 2021), respectively.

*2.6. Statistical Analysis*

Significant differences in the relative abundance of the microbial community members (phylum and class levels), pathways, and genes encoding N cycling components between the two ponds were determined by the Wilcoxon rank-sum test. The beta diversities of bacterial communities were estimated based on Bray–Curtis distances, and hierarchical clustering analysis was implemented using the unweighted pair-group method with arithmetic mean (UPGMA). The difference in the abundance of the underlying genes between either "SC and SCC" or "water and sediment" metagenomes were calculated as a rate of increase or decrease to the abundance of the corresponding genes. Then, the abundance percentages of genes were estimated using the following equation:

$$\text{R gene} = \frac{(a - b)}{b} \times 100\%$$

where R genes (%) represent the gene rates involved in N and S cycling that were selected. When we compared SCC with SC, "a" represents the abundance of the corresponding genes in W_SCC/S_SCC and b represents the W_SC/S_SC. When we compared the sediment with water in SC and SCC, "a" represents the abundance of the corresponding genes in the sediment of SC/SCC and "b" represents in those in the water of SC/SCC. To identify differences in the relative abundances of N cycling genes correlated with environmental factors, Spearman correlations were determined to identify the relationships between gene abundance and environmental factors in the R package psych. To determine the relative importance of gene sorting and dispersal limitation, forward stepwise regression was performed in the R package packfor.

## 3. Results

### 3.1. Microbial Community Composition in Water and Sediment

The microbial communities of the water in SC and SCC were primarily composed of members of the dominant phyla *Cyanobacteria*, *Actinobacteria*, *Proteobacteria*, *Bacteroidetes*, and *Verrucomicrobia* (Figure 1A) and the dominant classes *Cyanobacteria*, *Alphaproteobacteria*, *Actinobacteria*, *Bacteroidia*, and *Acidimicrobiia* (Figure 1B). There was no significant difference between SC and SCC in the microbial community members of the dominant phyla and classes.

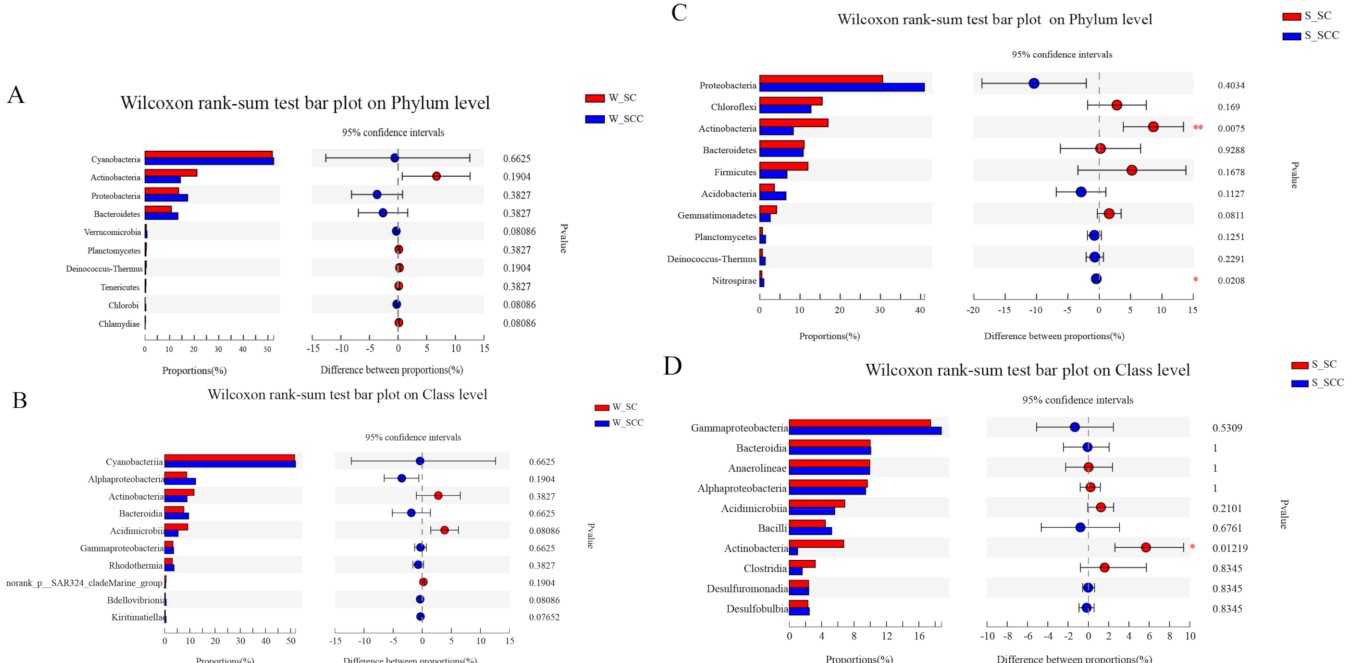

**Figure 1.** The differences in the microbial community composition in water and sediment in SC and SCC. Notes: (**A**) SC vs. SCC systems at the phylum level in water, (**B**) SC vs. SCC systems at the class level in water, (**C**) SC vs. SCC systems at the phylum level in sediment, and (**D**) SC vs. SCC systems at the class level in sediment. Asterisks (∗) indicate significant differences compared with the SC (Wilcoxon rank-sum test, $p < 0.05$).

The microbial communities in the sediment of SC and SCC were primarily composed of members of the dominant phyla *Proteobacteria*, *Chloroflexi*, *Actinobacteria*, *Bacteroidetes*, and *Firmicutes* (Figure 1C) and the dominant classes *Gammaproteobacteria*, *Bacteroidia*, *Anaerolineae*, *Alphaproteobacteria*, *Acidimicrobiia*, *Bacilli*, and *Actinobacteria* (Figure 1D). The relative abundance of *Actinobacteria* (phylum and class) was significantly lower in the sediment of SCC than in that of SC ($p < 0.05$). Furthermore, the microbial community composition was different between the water and sediment.

### 3.2. Microbial Community β Diversity in Water and Sediment

The microbial community structure was significantly different in SC and SCC (Figure 2). Four subgroups of pond samples were clearly identified based on the bacterial community composition, corresponding to the water of SC (W_SC), water of SCC (W_SCC), sediment of SC (S_SC), and sediment of SCC (S_SCC), but there was no clear separation between the sediment samples of the two ponds. These results indicated that the microbial communities of water and sediment were significantly different.

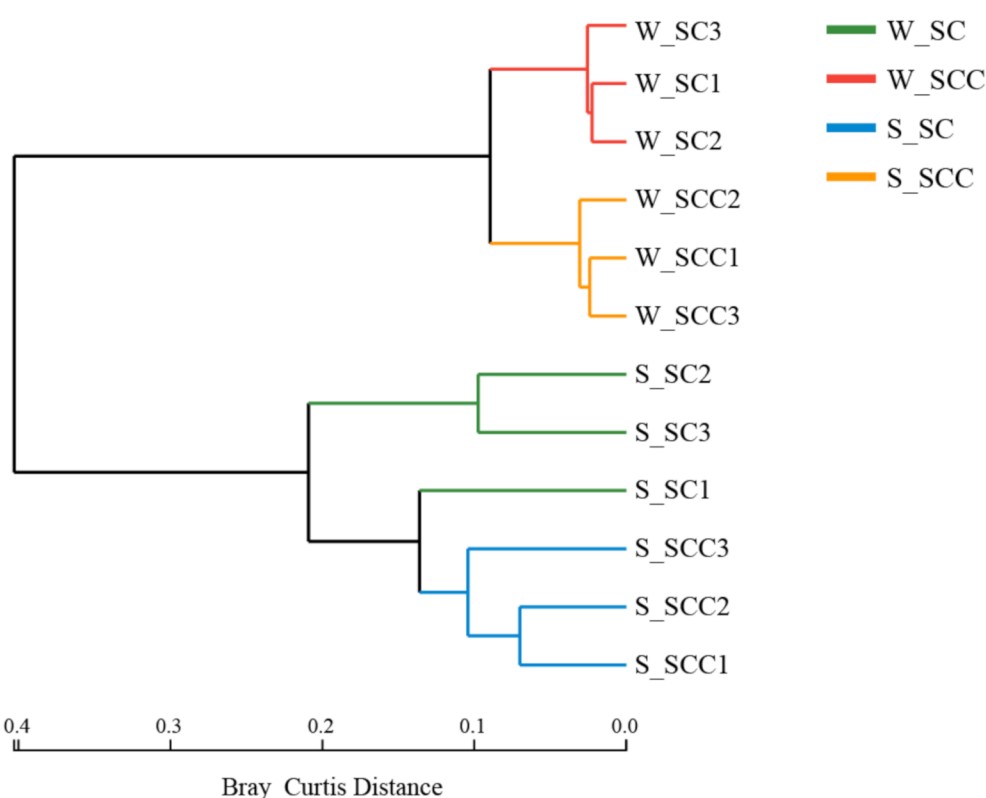

**Figure 2.** β diversity analysis of the SC and SCC ponds. Notes: S_SC—sediment of SC, S_SCC—sediment of SCC, W_SC—water of SC, W_SCC—water of SCC.

*3.3. Microbial Functional Profile-Based Feedback Direction*

The functions identified in SC and SCC were mainly consistent with metabolism (e.g., global and overview maps, amino acid metabolism, carbohydrate metabolism, and energy metabolism) (Figure 3A). Moreover, the top 15 most abundant KEGG level 3 pathways identified in SC and SCC and in both the water and sediment are presented in Figure 3B. In water, SCC showed a significantly higher relative abundance of metabolic pathways, microbial metabolism in diverse environments, glyoxylate and dicarboxylate metabolism, and the two-component system than SC ($p < 0.05$). In sediment, SCC showed a significantly lower relative abundance of oxidative phosphorylation and ABC transporters than SC ($p < 0.05$, Figure 3(B-2)). In addition, both SC and SCC showed significant differences in metabolic pathways between the sediment and water. The sediment showed significant increases in pathways involved in carbohydrate metabolism, cellular community prokaryotes, energy metabolism, membrane transport, and signal transduction relative to water ($p < 0.05$, Figure 3(B-3,B-4)), whereas metabolic pathways, the biosynthesis of secondary metabolites, and the biosynthesis of amino acids were significantly decreased in sediment relative to water ($p < 0.05$, Figure 3(B-3,B-4)).

*3.4. Expression of N Cycling Genes in Microbes*

The identified genes involved in the uptake and biosynthesis of N and S compounds are presented in Figure 4. Genes involved in the uptake and biosynthesis of N compounds were generally more abundant in SCC than in SC in water, but were generally more abundant in SC than in SCC in sediment (Figure 4A). In water, there was no significant difference in nitrate, ammonium binding, and acquisition genes between SC and SCC ($p > 0.05$). In sediment, an assimilatory nitrate reductase gene (*nasA*) and genes specific to complete nitrification (*narG*), genes specific to denitrification (*norB*) and a dissimilatory nitrate reduction gene (*nirB*) presented significant differences between SC and SCC ($p < 0.05$), and the relative abundance of these genes was significantly lower in SCC than SC by 82%,

23%, 45%, and 70%, respectively. However, genes related to S-dependent denitrification and respiratory DNRA, the sulfur oxidizer gene *soxB,* and a gene encoding both a sulfur oxidizer and sulfate reducer (*aprA*) were not significantly different between SC and SCC in sediment or water (*p* > 0.05).

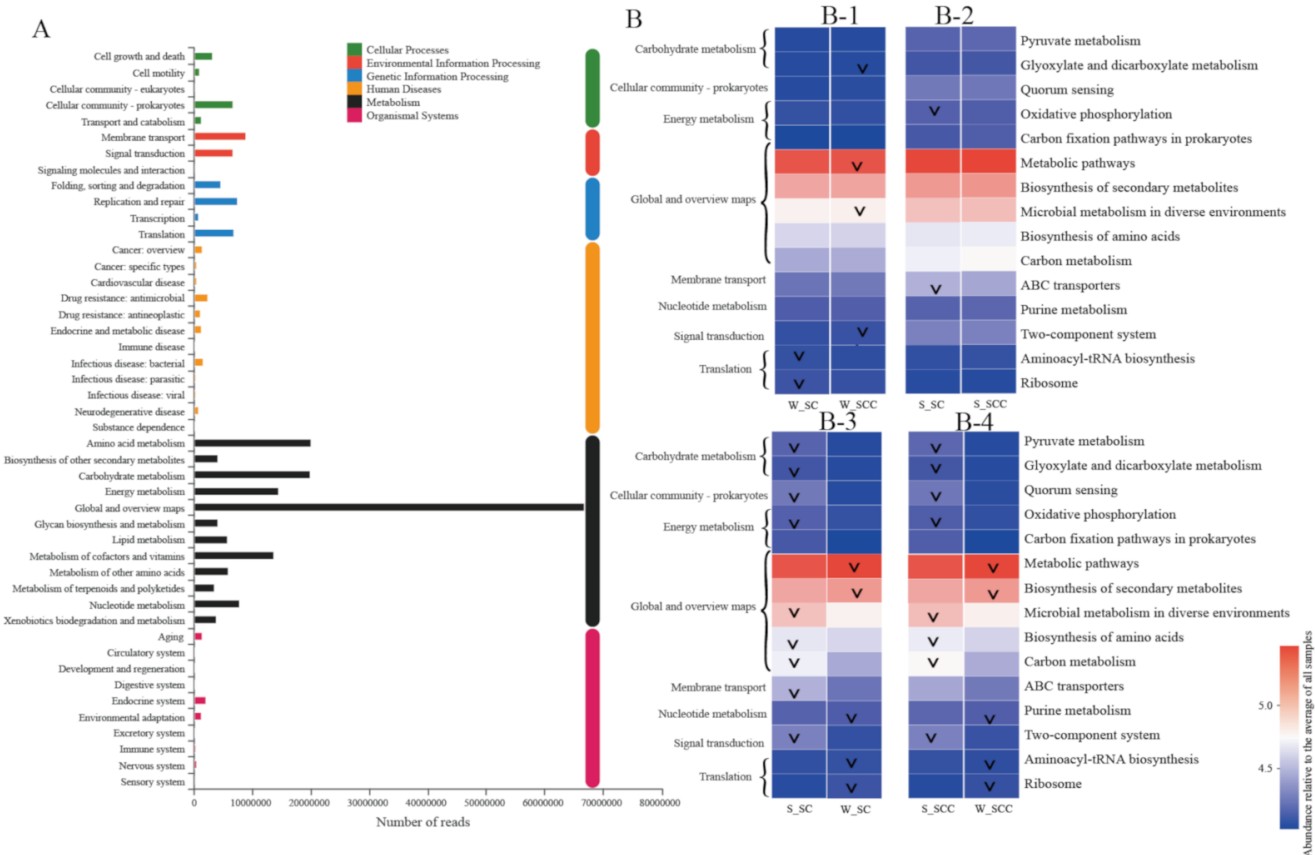

**Figure 3.** Pathway abundance in SC and SCC pond samples. Notes: (**A**) The dominant pathways in both SC and SCC. (**B**) Differences in pathway abundance based on KEGG level 3, (**B-1**) pathway differences between SC and SCC in water, (**B-2**) pathway differences between SC and SCC in sediment, (**B-3**) pathway differences between water and sediment in SC, and (**B-4**) pathway differences between water and sediment in SCC. "v" indicates that the abundant is significantly higher than that of the other group (*p* < 0.05).

The differences in the relative abundance of N cycling genes between water and sediment are presented in Figure 4B. Genes involved in the uptake and biosynthesis of N compounds were generally more abundant in sediment than in water in both SC and SCC. Relative to water, sediment showed more significant abundance of N acquisition genes responsible for several steps involved in denitrification, nitrification, and DNRA including the *nirK*, *nirS*, *norB*, *nosZ*, *nirB*, and *nrfA* genes and genes specific to S-dependent ANAM-MOX sulfate reductase (*dsrA*) (*p* < 0.05). However, sediment showed significantly lower expression of genes specific to assimilatory nitrate reduction (*narB* and *nirA*) (*p* < 0.05).

### 3.5. Drivers of the Differences in the Relative Abundance of N Cycling Genes

Distinct differences in the relative abundance of N cycling genes in microbial communities were observed between SC and SCC and between the water and sediment. Hence, the potential intrinsic factors driving the differences in the relative abundances of these genes were further explored to identify environmental properties (Tables S4 and S5) that affect the differences in the relative abundances of N cycling genes. The results for these genes were correlated with the environmental properties of the water and sediment of SC and

SCC (Figure 5A). In water, salinity, pH, and *S. constricta* density were significant positive predictors of the differences in the relative abundance of most N cycling genes ($p < 0.05$). In sediment, P_$NH_4^+$ and *S. constricta* density were significant negative predictors of the differences in the relative abundance of most N cycling genes.

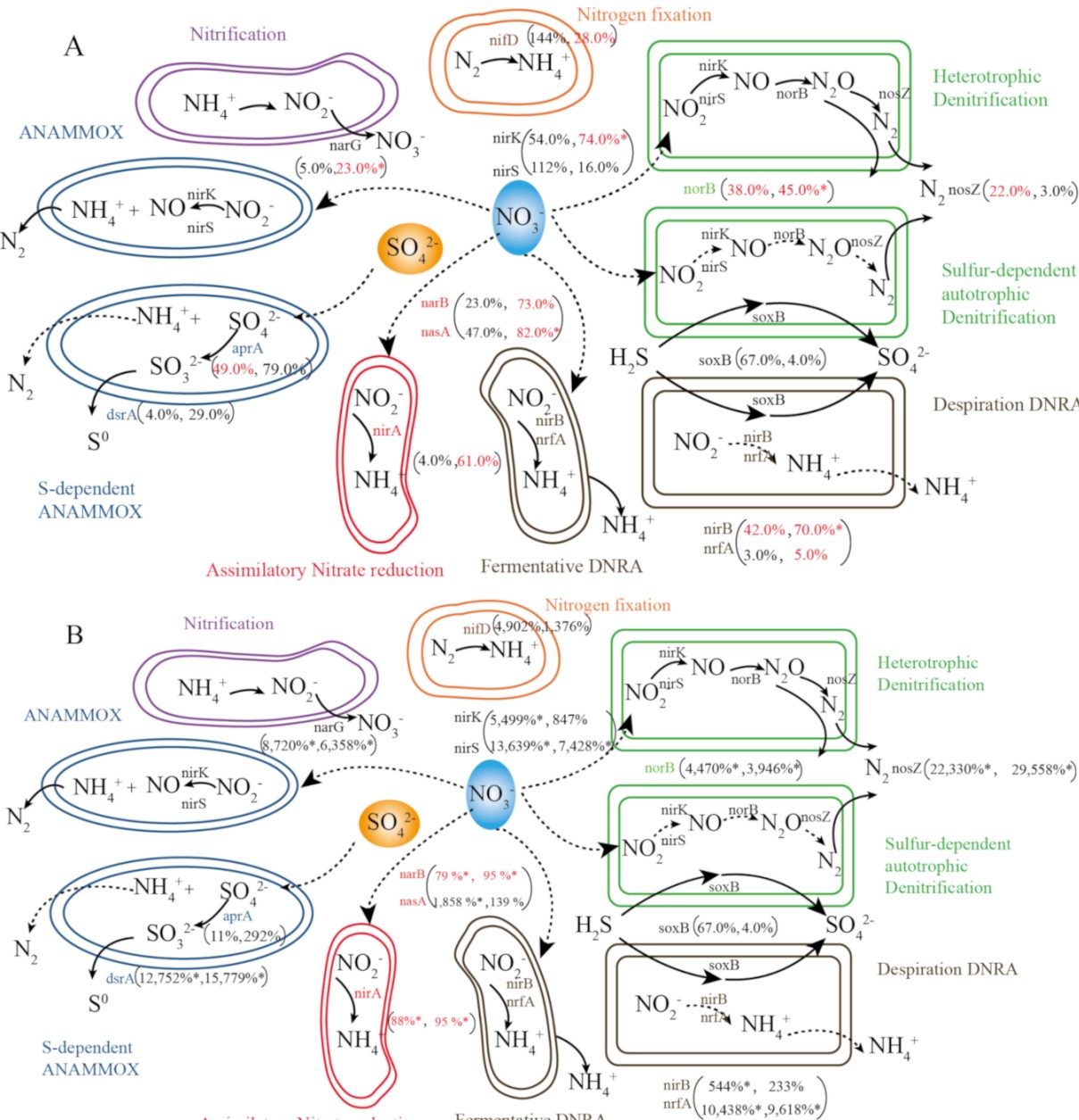

**Figure 4.** A diagram representing selected N cycling pathways and the differences in abundance of the underlying genes. Notes: (**A**) Differences in genes underlying N cycling between the SC and SCC metagenomes. Percentages are ordered as follows: SCC vs. SC in water and SCC vs. SC in sediment. (**B**) The differences in underlying genes between the water and sediment metagenomes. Percentages are ordered as follows: sediment vs. water in SC and sediment vs. water in SCC. Percentages indicated in black or red represent an increase or decrease in the abundance of the corresponding genes, respectively. Percentages denoted with an asterisk (*) represent abundance changes that were statistically significant ($p < 0.05$).

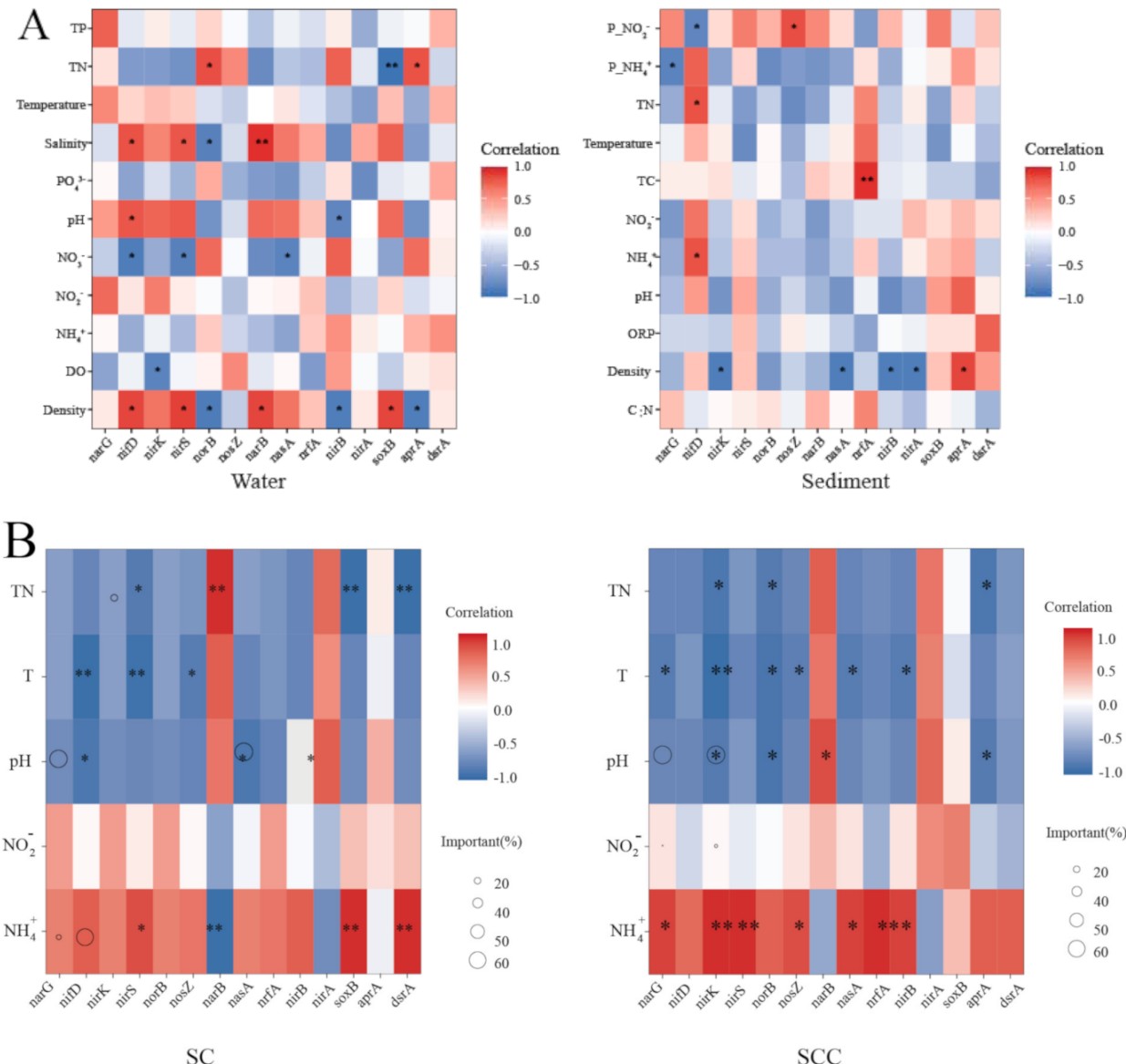

**Figure 5.** Contributions of environmental properties to differences in the relative abundances of N cycling genes. Notes: (**A**) The correlations of functional gene differences with environmental properties between SC and SCC in the water and sediment. (**B**) The correlations of functional gene differences with environmental properties between the water and sediment in SC and SCC and the identified major predictors. Circle size represents the importance of the variable. Colors represent Spearman correlations, and significance is indicated with asterisks (* *p* < 0.05; ** *p* < 0.01).

The environmental properties affecting the differences in the relative abundances of N cycling genes between water and sediment are presented in Figure 5B. pH was a significant negative predictor of differences in the relative abundances of most N cycling genes in both SC and SCC. $NH_4^+$ was a significant, positive predictor of differences in the relative abundances of most N cycling genes in SC and SCC. In addition, pH was a significant positive predictor for *narB* and *nirA*, and $NH_4^+$ was a significant negative predictor for *narB* and *nirA*. The environmental factors made different contributions to the differences in the relative abundances of most N cycling genes. $NH_4^+$ was the dominant contributor to differences in the relative abundance of *nifD* in SC, pH was the dominant contributor to differences in the relative abundances of *narG* and *nasA* in SC, and pH was also the dominant contributor to differences in the relative abundances of *narG* and *nirK* in SCC.

## 4. Discussion

The present study characterized the microbiomes of *P. japonicus-P. trituberculatus* and *P. japonicus-P. trituberculatus-S. constricta* ecosystems, which are commonly found in the middle and eastern areas of China. The dominant members of the systems assigned to phylum *Cyanobacteria* and class *Cyanobacteria* in water and to phylum *Proteobacteria* and class *Gammaproteobacteria* in sediment were in accordance with a previous report [1,40]. These results indicated that the richest phyla and classes were *Cyanobacteria* and *Cyanobacteria*, respectively, in water, and *Proteobacteria* and *Gammaproteobacteria*, respectively, in sediment in both SC and SCC. Members of *Actinobacteria* have been proven to be involved in the decomposition of organic matter and C fixation in the environment [2], and the decreased abundance of *Actinobacteriota* and *Actinobacteria* in the sediment of SCC indicated that bioturbation and filtration may reduce the accumulation of organic matter in sediment under the coculture of *S. constricta* in shrimp-crab ponds. Furthermore, the results indicated that the compositions of the microbial communities largely differed between water and sediment. These results highlight the disparity of the phylum and class composition in the water and sediment microbial communities and further support the proposed importance of habitat specificity in structuring their composition.

The response of the microbiome to the coculture of *S. constricta* in shrimp-crab ponds and different environments involved a re-accommodation of microbial community β-diversity. The separation of the microbial community may indicate the functions of rare species and revealed the stability of the core microbial community [41,42]. In this study, the results indicate that the clear separation of SC and SCC in water samples would mainly be affected by changes in rare species. The unclear separation of SC and SCC may indicate the stability of the core microbial community of sediment. Rather, the clear separation of water and sediment reflects the compositional stability of the core microbial community of water and sediment. These results indicated the compositional stability of the core bacterial community in water and sediment. Rare species may play an important role in the stability of the microbial community of the sediment. Thus, assessing changes in gene expression at the metagenomic level appears to be necessary to further understand the pond microbiome.

The characteristics of the pond macroecological environment (e.g., bioturbation) can encourage the interaction of chemical species, providing niches for microbial metabolic activity to catalyze reactions such as those involving C, N, and S [43,44]. In this study, SCC showed significant increases in the glyoxylate and dicarboxylate metabolism and two-component systems in water relative to SC. Glycoxylate and dicarboxylate metabolism are typical C metabolic pathways, and two-component systems are essential for C metabolism [45]. The above results indicated that *S. constricta* may mainly cause a shift C cycling in the water of shrimp-crab ponds. Although the dominant pathways showed no significant variation, relatively rare pathways such as oxidative phosphorylation and ABC transport were decreased in the sediment of SCC compared to that of SC. Oxidative phosphorylation is involved in ATP synthesis, and ABC transporters including many types of prokaryotic-type ABC transporters can carry out mineral and organic ion transport such as that of nitrate/nitrite [46,47]. The decreased abundances of these pathways in the sediment of SCC indicated that the coculture *S. constricta* in shrimp-crab ponds may limit the activity of energy metabolism and have the main effect of shifting N metabolism in sediment. In addition, the relative abundance of the main metabolic pathways in sediment was significantly higher than that in water, indicating that sediment may show higher nutrient metabolism activity.

Studies have indicated that, among the biological or physicochemical indexes (e.g., pH and N concentrations), gene relative abundance integrated with information on both the environmental history and recent process activity was the best predictor of element cycling and an adequate predictor of the associated enzyme activity [26,48,49]. In this study, SCC showed higher relative abundances of key genes related to the main N cycle in water than SC, indicating that coculture with *S. constricta* in shrimp-crab ponds promotes most N processes in water. In sediment, the significant decreases in the key

genes *narG, norB, nasA*, and *nirB* in SCC relative to SC indicated declines in nitrification, denitrification, assimilatory nitrate reduction, and DNRA. In addition, *soxB* is the key gene in autotrophic denitrification and respiratory DNAR [50], and the observed increase in its relative abundance may indicate that autotrophic denitrification is the main mechanism of denitrification and that respiratory DNAR is the main mechanism of DNAR. Autotrophic denitrification usually occurs when organic C substrates are insufficient; respiratory DNAR is a chemolithoautotrophic process that reduces nitrate and oxidizes S compounds [20]. The above results indicated that *S. constricta* may reduce the accumulation of organic C and promote chemolithoautotrophic processes in sediment. In addition, the coculture of *S. constricta* in shrimp-crab ponds may stimulate N cycling in water, but limit that in sediment.

There are significant differences in biogeochemical cycles between water and sediment; in the ocean, for example, reductions in the C flux primarily originate in water, whereas reductions in the sulfur flux primarily originate within sediments [43]. In this study, in addition to the specific genes related to assimilatory nitrate reduction, there was a larger increase in the key genes involved in N cycling in sediment than in water in both SC and SCC, suggesting a more active process of N cycling in sediment. Through the assimilatory nitrate reduction process, inorganic N can be transformed into microbial biomass [26], and the significant decrease in this process observed in sediment indicated that sediment is not well suited to the growth of microbes, which is favored by acquiring N in biomass. These results revealed that reduced N is primarily formed within the sediments in shrimp IMTA ponds, with the exception of assimilatory nitrate reduction. Thus, further study of the drivers of the variation and the dissimilarity of the microbial community and differences in N cycling genes is urgently needed.

Microbial community function is affected by the type of cultured species and diversity that can affect the microbial N decomposition potential by supplying energy and materials (e.g., nutrient waste) to microbes [5]. Water properties such as salinity and pH may favor the growth of certain microbes, showing differences in N use efficiency [32,51]. Salinity differences can affect external osmolarity around the cell and thereby stimulate cellular osmoregulation mechanisms, which affect microbial community function [52]. pH can impose a physiological constraint on microbes by altering the competition or reducing the net growth of some taxa [53]. Considering the increased abundance of N cycling genes in the shrimp-crab-clam pond together with the significant positive contribution of water salinity and pH to the differences in the relative abundance of N cycling genes between SC and SCC, the above results indicate that the coculture of *S. constricta* in a shrimp-crab pond improved the salinity and pH conditions for osmoregulation, thereby exerting positive feedback on microbes and creating better physiological conditions for microbial N cycling. Sediment properties including pH, $NH_4^+$, and temperature regulate enzyme activities involved in N metabolism [42]. In this study, P_$NH_4^+$ showed significant negative factors affecting the relative abundance of N cycling genes in sediment. Considering the significantly lower relative abundance of N cycling genes in SCC than in SC, the results indicated that coculture with *S. constricta* in shrimp-crab ponds may decrease the bioavailability of nitrogen, especially in the form of $NO_2^-$ and $NH_4^+$.

pH impacts the bioavailability of multiple elements as ammonia availability is highly pH dependent [53]. In this study, pH (as a negative predictor) and $NH_4^+$ (as a positive predictor) affected the relative abundances of most N cycling genes between the water and sediment. pH was a significant positive predictor of *narB* and *nirA* abundance, and $NH_4^+$ was a significant negative predictor of *narB* and *nirA* abundance in the water and sediment. Together with the finding that sediment showed a higher abundance of N cycling genes than water, these results indicate that sediment pH may increase the concentration of $NH_4^+$ by limiting assimilatory nitrate reduction because this process can assimilate $NH_4^+$ into the biomass of microbes; therefore, limiting this process could increase the concentration of $NH_4^+$, which is important for N cycling. Hence, the reduced N flux originates primarily within the sediment of shrimp IMTA ponds.

## 5. Conclusions

In this study, the structures and N cycling genes of the microbial communities of two typical shrimp IMTA ponds were revealed. The results indicated that coculture with *S. constricta* in shrimp-crab ponds contributed little to the microbial taxonomic patterns but significantly affected N cycling. N cycling increased in water but declined in sediment. In addition, the reduced N flux originated primarily within the sediments of the shrimp IMTA ponds, with the exception of assimilatory nitrate reduction. Environmental factors such as pH and $NH_4^+$ played important roles in influencing microbial community N cycling in the shrimp IMTA ponds. Understanding the adaptation of microbial community potential functions to water, sediment, and their influencing factors may provide new insights into the regulation of aquaculture N.

**Supplementary Materials:** The following supporting information can be downloaded at: https://www.mdpi.com/article/10.3390/jmse10020171/s1. Figure S1: The SC and SCC ponds are located in Changbai Bay (122.04° E, 30.19° N). The unfilled dots represent the water and sediment sampling sites. Table S1: Pond seed stocking situation. Table S2: PCR parameters. Table S3: PCR system. Table S4: Environmental factors in water from SC and SCC. Table S5: Environmental factors in sediment from SC and SCC.

**Author Contributions:** Q.L. performed the experiments and wrote the manuscript. J.L. and Y.X. performed data curation and formal analysis, H.S. carried out the writing, review, and supervision. All authors have read and agreed to the published version of the manuscript.

**Funding:** This work is supported by National Key R&D Program of China, no. 2019YFD0900402. The authors are grateful to the referees for their helpful comments on the manuscript.

**Institutional Review Board Statement:** Not applicable.

**Informed Consent Statement:** Not applicable.

**Data Availability Statement:** The data presented in this study are available on reasonable request from the corresponding author.

**Acknowledgments:** The authors greatly appreciate the anonymous reviewers for their careful work and thoughtful suggestions that substantially improved this paper.

**Conflicts of Interest:** The authors declare no conflict of interest.

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
