# Peer review of "Metagenomic Insights into the Structure of Microbial Communities Involved in Nitrogen Cycling in Two Integrated Multitrophic Aquaculture (IMTA) Ponds"

_jmse, doi:10.3390/jmse10020171_

Round 1

Reviewer 1 Report

Title:

    Metagenomic insights into the structure of microbial communities involved in nitrogen cycling in two integrated multitrophic aquaculture (IMTA) ponds

Recommendation:

 Accept after minor revision.

Comments:

This study evaluated the structure of microbial communities involved in nitrogen cycling in two integrated multitrophic aquaculture ponds by using metagenomic analysis. The subject is relevant and consistent with the aims and scopes of the journal. In my opinion, this study provided a new insight into the structure of microbial communities in integrated multitrophic aquaculture ponds. One comment and suggestion is offered below with the intent to assist the author in improving the manuscript.

  1. As mentioned in lines 213 to 236, sediment showed significant increases in pathways involved in carbohydrate metabolism, cellular community prokaryotes, energy metabolism, membrane transport and signal transduction relative to water, whereas metabolic pathways, the biosynthesis of secondary metabolites, and the biosynthesis of amino acids were significantly decreased in sediment relative to water. However, in Fig. 3B-3 and 4, I can't see the trend described in the article. Please provide a more complete interpretation in the text.

Author Response

Response 1: We thank the reviewer for this comment. The experimental results were not well shown in Figure 3B before. We have redrawn Figure 3B to show the experimental results more clearly.

Reviewer 2 Report

Summary

The authors presented a comparative and descriptive study between two types of integrated multitrophic aquaculture (IMTA) shrimp ponds. They based their work on the fact that the nitrogen is one of the key nutrients limiting the production in the aquaculture ecosystems and the crucial role of microbes in nitrogen cycling. To address this question, first, they evaluated the microbial community composition of each ponds differentiating water and sediment samples. Secondly, they performed a metagenomics sequencing analysis in order to identify the genes related to the nitrogen cycling. They found that the nitrogen flux originated primarily in the sediments; however, the presence of the clam in the pond shifted the nitrogen cycling to the water. When analyzing the impact of the environmental factors, they found that pH and NH4+ were the parameters influencing mostly the nitrogen cycling in the microbial communities, as negative and positive predictors, respectively. In light of the results obtained, this study unveiled factors to take into account for the regulation of microbial communities in IMTA ecosystems.

The rationale followed by the authors in the design and performance of the experiments is accurate to answer the proposed questions. Methods, tools, software and reagents are mostly described in detail and applied statistics are mentioned along with the significance values. The databases where the sequencing results were deposited are also listed together with the accession numbers. The results are mostly well presented, some minor suggestions are made in the comments below.

The authors are asked to please address the following comments.

  • According to the results presented in the Figure 1, in the legend of that figure, lines 208-209, I think the significant differences are between the two types of ponds (SC vs. SCC) instead of the comparison with the control. In case that the original text is correct, can the authors explain what is considered as the control?
  • In Figure 2: specify the title of the scale in the bottom of the figure, I think should be distance but would be better the authors to write the correct description.
  • In the left axis of the Figure 3A the pathway names and the color bars are shifted, could they remake it so that the correlation between the pathways and the abundance can be better displayed?

Figure 3B: heat map scale is covered by the text of the pathway’s name. Can the authors specify what do the colors represent in the heat map scale?

  • Related to the results in Figure 4. Could the authors add in the methodology a description of how they calculate the percentages? Is the abundance change calculated against some reference genes or it is just the comparison between pair conditions mentioned in the legend?
  • For the description of the results in Figure 5A, when the authors mentioned that in water the TN is a positive predictor and P_NO2- is a negative predictor in sediments, seems not so clear by looking at the figure. Can they explain it a little more or consider removing it from the sentence in lines 286, 288 and 402?
  • Figure 5B: the two upper graphs seem redundant and not contributing in the explanation of the lower graphs.
  • Line 74: typo microbiome
  • Line 323: revise word order in the beginning of the sentence
  • Line 332: indicate that the clear separation of SC and SCC is in water samples
  • Line 365: seems incomplete sentence, … in SCC relative to SC indicated

Author Response

Response 1: We thank the reviewer for this comment. We have replaced “control” with “SC” in the revised manuscript (line 220).

Response 2: We thank the reviewer for this comment. The title of the scale has been added in the bottom of the Figure 2 in the revised manuscript.

Response 3: We thank the reviewer for the suggestion. We have redrawn Figure 3A and added the heat map scale in Figure 3B.

Response 4: We thank the reviewer for the suggestion. We have added a description in the methodology about the percentages (lines180-189). The abundance change calculated between pair conditions mentioned in the legend.

Response 5: We thank the reviewer for this comment. We have deleted TN and P_NO2- in the revised manuscript.

Response 6: We thank the reviewer for this comment. We have deleted the two upper graphs in Figure 5B.

Response 7: We thank the reviewer for this comment. We have changed the “microbe” to “typo microbiome” in the revised manuscript (line74).

Response 8: We thank the reviewer for this correction. We have revised word order in the revised manuscript (line 337).

Response 9: We thank the reviewer for this comment. We have added the sentence to the revised manuscript (lines 346-347).

Response 10: We thank the reviewer for this correction. We have revised the sentence in the revised manuscript (line 380).
